# 9/11 Residential Exposures: The Impact of World Trade Center Dust on Respiratory Outcomes of Lower Manhattan Residents

**DOI:** 10.3390/ijerph16050798

**Published:** 2019-03-05

**Authors:** Vinicius C. Antao, L. Lászlo Pallos, Shannon L. Graham, Youn K. Shim, James H. Sapp, Brian Lewis, Steven Bullard, Howard E. Alper, James E. Cone, Mark R. Farfel, Robert M. Brackbill

**Affiliations:** 1Division of Toxicology and Human Health Sciences, Agency for Toxic Substances and Disease Registries, 4770 Buford Highway NE, Mail Stop F-58, Atlanta, GA 30341, USA; antaov@hss.edu (V.C.A.); Laszlo.Pallos@cdc.HHS.gov (L.L.P.); buu9@cdc.gov (S.L.G.); yak3@cdc.gov (Y.K.S.); ozs1@cdc.gov (J.H.S.); bkl9@cdc.gov (B.L.); asz3@cdc.gov (S.B.); 2Center for the Advancement of Value in Musculoskeletal Care, Hospital for Special Surgery, 535 East 70th Street, New York, NY 10021, USA; 3Department of Health and Mental Hygiene, World Trade Center Health Registry, 125 Worth St, New York, NY 10013, USA; halper@health.nyc.gov (H.E.A.); jcone@health.nyc.gov (J.E.C.); mfarfel@health.nyc.gov (M.R.F.)

**Keywords:** WTC attack, respiratory symptoms, lower Manhattan residents, cleaning practices

## Abstract

Thousands of lower Manhattan residents sustained damage to their homes following the collapse of the Twin Towers on 11 September 2001. Respiratory outcomes have been reported in this population. We sought to describe patterns of home damage and cleaning practices in lower Manhattan and their impacts on respiratory outcomes among World Trade Center Health Registry (WTCHR) respondents. Data were derived from WTCHR Wave 1 (W1) (9/2003–11/2004) and Wave 2 (W2) (11/2006–12/2007) surveys. Outcomes of interest were respiratory symptoms (shortness of breath (SoB), wheezing, persistent chronic cough, upper respiratory symptoms (URS)) first occurring or worsening after 9/11 W1 and still present at W2 and respiratory diseases (asthma and chronic obstructive pulmonary disease (COPD)) first diagnosed after 9/11 W1 and present at W2. We performed descriptive statistics, multivariate logistic regression and geospatial analyses, controlling for demographics and other exposure variables. A total of 6447 residents were included. Mean age on 9/11 was 45.1 years (±15.1 years), 42% were male, 45% had ever smoked cigarettes, and 44% reported some or intense dust cloud exposure on 9/11. The presence of debris was associated with chronic cough (adjusted OR (aOR) = 1.56, CI: 1.12–2.17), and upper respiratory symptoms (aOR = 1.56, CI: 1.24–1.95). A heavy coating of dust was associated with increased shortness of breath (aOR = 1.65, CI: 1.24–2.18), wheezing (aOR = 1.43, CI: 1.03–1.97), and chronic cough (aOR = 1.59, CI: 1.09–2.28). Dusting or sweeping without water was the cleaning behavior associated with the largest number of respiratory outcomes, such as shortness of breath, wheezing, and URS. Lower Manhattan residents who suffered home damage following the 9/11 attacks were more likely to report respiratory symptoms and diseases compared to those who did not report home damage.

## 1. Introduction

The terrorist attacks in New York City on 11 September 2001 (9/11) led to the destruction of the World Trade Center (WTC) Twin Towers and six other adjacent buildings. The collapse of the two towers released a massive cloud of dust and debris that damaged surrounding buildings within and around the WTC complex, a 16-acre area that was subsequently called Ground Zero [1]. The plume of dust reached an altitude of 1500 meters and, depending on meteorological conditions, extended to lower Manhattan and the New York and New Jersey metropolitan areas. More than 100,000 µm per cubic meter of particles were estimated to be present in the air during the first few minutes after the collapse of each building [2].

The contents of settled WTC dust have been extensively analyzed and were characterized as a mixture of cement dust, glass fibers, asbestos, lead, polycyclic aromatic hydrocarbons, polychlorinated biphenyls, organochlorine pesticides, and polychlorinated furans and dioxins [1,3,4,5]. Indoor dust was primarily comprised of inhalable particles (<53 µm). In contrast, most outdoor dust was primarily comprised of larger particles [6].

Approximately 25,000 lower Manhattan residents were physically and emotionally affected by this disaster [7]. Two months after the attacks, air quality and surface dust were among the main concerns of lower Manhattan residents [7]. Most WTC-related residential exposures were caused by re-suspension of settled indoor dust [2] during cleaning efforts or in poorly cleaned environments. This is a large motivator for our analyses of cleaning practices.

A number of studies demonstrated the deleterious health effects of household exposures in the context of the WTC attacks. Most of them were relatively small case-control studies [8,9,10,11]. Increased asthma prevalence was previously reported among WTC Health Registry (WTCHR) enrollees who experienced a heavy layer of dust in their homes [12]. Acute and chronic exposures were also associated with lower respiratory symptoms among WTCHR participants [13]. However, prior studies, specifically Lin et al. [8] and Maslow [13], had limited samples of residents (n = 1317; n = 479, respectively). Maslow [13] did not evaluate associations between exposures and specific respiratory symptoms. Although Lin et al. [8] did assess lower and upper respiratory symptoms, they did not include diagnosed conditions such as asthma or chronic obstructive pulmonary disease.

With a large sample of residents in lower Manhattan and a comprehensive set of outcomes, the present study describes patterns of home damage and cleaning practices, as a surrogate for household exposures since measures of exposure in homes is limited. We initially looked at geospatial relationships between Ground Zero location and exposure and health outcomes. We then conducted multivariate logistic regression analyses to find potential associations between those variables among a large sample of WTCHR respondents.

## 2. Methods

The WTCHR monitors the health of people exposed to the 9/11 WTC disaster through periodic health surveys and is housed within the New York City Department of Health and Mental Hygiene (NYCDHMH). The WTCHR is the largest effort in the United States to monitor health after a disaster of this kind and includes data on 68,444 adults, including lower Manhattan residents, rescue and recovery workers, and building occupants and passers-by. More details on the WTCHR can be found elsewhere [14] and online at https://www1.nyc.gov/site/911health/about/wtc-health-registry.page.

For this report, data were derived from WTCHR Wave 1 (W1) survey (baseline), collected between 9/2003 and 11/2004, and Wave 2 (W2) survey (first follow-up), collected between 11/2006 and 12/2007. We included W1 and W2 participants 18 years and older, whose primary residence on September 11th was located in lower Manhattan, south of Canal Street. Residents who performed rescue and recovery work were excluded from this analysis.

The Centers for Disease Control and Prevention (CDC) and NYCDHMH institutional review boards approved the Registry protocols. A Federal Certificate of Confidentiality was obtained, as was verbal informed consent.

### 2.1. Environmental and Household Exposures

In the W1 Survey, respondents were asked if they were outdoors after the towers’ collapse and to report their location when they first encountered the dust cloud (closest cross street intersection, closest landmark, or closest subway stop) [14]. The W2 survey information was used to categorize dust exposures into “intense”, “some”, and “none”. The “intense” dust exposure was defined as having been in the dust cloud on 9/11 and reporting at least one of five experiences: being unable to see more than a few feet; having difficulty walking or finding one’s way; trouble finding shelter; being covered with dust; or not being able to hear. The “some” category consisted of those who had reported being in the dust and debris cloud at Wave 1 but who did not experience intense exposure and “none” were those who reported no dust cloud exposure at all. These categories of exposure to the dust cloud (1 = none, 2 = some, or 3 = intense) were combined with reported location at the time of first encounter with the dust cloud (reported at W1) to produce Figure 1 for the geospatial analysis.

Wave 2 also included detailed questions about conditions inside the homes after 9/11 as well as cleaning practices and replacing of household items. WTCHR participants were asked about the presence of “a fine coating of dust on surfaces”, “a heavy coating of dust on surfaces (so thick one couldn’t see what was underneath)”, “broken window(s)”, “damage to home or furnishings”, and “debris from the disaster”. In addition, enrollees answered whether or not they had personally “cleaned ventilation ducts”, “cleaned with a damp cloth or wet mop or wet sponge”, “used a vacuum (with or without a high-efficiency particulate air or HEPA filter)”, and “dusted or swept without water.” Moreover, enrollees were inquired whether or not they replaced “carpet or rugs”, “furniture (replaced or re-upholstered)”, “drapes, blinds or curtains”, and “air conditioners” as a result of the 9/11 events.

### 2.2. Health Outcomes

Outcomes of interest were self-reported respiratory symptoms and self-reported physician diagnoses of respiratory diseases, asked at both W1 and W2 surveys. Symptoms included shortness of breath, wheezing, persistent chronic cough, and upper respiratory symptoms, first occurring or worsening after 9/11 and present at W2. Respiratory diseases comprised asthma or reactive airways dysfunction syndrome (RADS) and chronic obstructive pulmonary disease (COPD), first diagnosed after 9/11 and present at W2.

### 2.3. Geospatial Analyses

Initially, to explore potential geospatial relationships between Ground Zero location and exposures and health outcomes, we mapped exposures and health outcomes of WTCHR resident enrollees in lower Manhattan using ArcGIS version 10.2 (ESRI, Redlands, CA, USA). These variables were mapped at the census tract level of geography using 2000 Census tract boundaries. For this part of the analysis, “yes” responses for any of the health outcomes variables were summed per census tracts (Boolean OR, not arithmetic sum). Similarly, exposure variables (fine coating of dust on surfaces, heavy coating of dust on surfaces, broken windows, damage to home or furnishings, and debris from the disaster was present) were also grouped and categorized as “yes” or “no” and mapped by census tract totals. The number of enrollees within each census tract was used as the denominator of the proportions represented on the maps. In addition, point values of dust cloud exposure were modeled using inverse distance weighted (IDW) interpolation to create a surface representation of the dust levels throughout lower Manhattan. IDW uses measured values (point locations) to predict values for unmeasured surrounding areas. 

### 2.4. Statistical Analyses

We examined demographic characteristics, household exposures resulting from the 9/11 attacks, and cleaning practices among the study population using descriptive statistics (means, standard deviations, and proportions). We also calculated the prevalence (%) of self-reported respiratory symptoms and respiratory diseases. The relationship between cleaning practices and conditions inside the homes after 9/11 was assessed using multivariate logistic regression analyses adjusted for age, race, education, income, sex. Analyses were also controlled for “priority group”, a stratification of registrants into higher and lower priority groups for the purposes of targeting and outreach, which to a degree reflects a resident’s distance from the WTC site. Priority group 1 included residents as of 9/11 at addresses located south of Chamber Street; Group 2 included residents located on or north of Chamber but south of Canal Street; a third resident group, Group 0, was defined upon recognition of the inclusion of respondents living on or north of Canal in zip codes overlapping the Canal boundary or in other NYC boroughs. Also, the analyses were controlled for any exposure outside, to the dust cloud.

Each of the cleaning practices (dusted or swept without water; cleaned with damp cloth, sponge, or mop; used a vacuum to clean; and cleaned ventilation ducts) was modeled using logistic regression. Based on medical considerations and plausibility of real relationships being present, the list of all variables was narrowed to 17 potential explanatory variables for cleaning practices using ‘purposive’ or *a priori* selection. The predictors or covariates included five home exposures (fine coating of dust, heavy coating of dust, broken windows, damage in the home or furnishings, presence of debris in the home). The covariates also included four replacement behaviors (replaced AC, carpets, drapes, furniture). Lastly, each model contained terms for eight confounders, as listed in Section 3.2 (Table 3).

The cleaning practices themselves, other than the cleaning behavior specified outcome, are also modeled as predictors. These other cleaning practices were conditioned on the presence of dust in the home. For example, “cleaned ventilation ducts” was modelled including the other three cleaning practices as predictors (i.e., dusted or swept without water; used a vacuum to clean; and cleaned with damp cloth, sponge, or mop). The covariate cleaning practices were considered only if reported in conjunction with the presence of either fine or heavy dust (e.g., vacuum and fine or heavy dust had to be “yes” for the derived covariate “vacuum” to be “yes”). We fitted the four logistic models with all 17 variables included. Adjusted odds ratios (aOR), and their 95% confidence intervals (95% CI), for each cleaning method are provided in Section 3.2 (Table 3). 

Each health outcome variable was modelled in a similar way. Based on medical considerations and likelihood of real relationships being present, the list of all covariates was narrowed to 21 explanatory variables for health outcomes (again, ‘purposive’ or *a priori* selection). Each of the six health outcomes was modeled using logistic regression with covariates as follows: (a)Five demographic variables (age at 9/11, race, sex, education, and income);(b)Resident priority group, indicating degree of registrant recruitment by area (partially a surrogate for location relative to the WTC site);(c)Dust cloud exposure category (intense, some, none);(d)Ever having smoked (at least 100 cigarettes in a lifetime);(e)Five variables of exposures in the home (broken windows, debris in the home, fine dust, heavy dust, and damage in the home);(f)Four variables containing data on behavior regarding replacement of various major items in the home (having replaced air-conditioning, carpeting, drapes, and furniture); and(g)Four variables for the presence of cleaning practices, which are conditioned on the presence of dust in the home.

Again, the covariate cleaning practices were considered only if reported in conjunction with the presence of either fine or heavy dust (e.g., vacuum and fine or heavy dust had to be “yes” for the derived covariate “vacuum” to be “yes”). Smoking status was queried as “Have (you) smoked at least 100 cigarettes in (your) entire life?” Adjusted odds ratios (aOR), and their 95% confidence intervals (95% CI), for each health outcome were computed for all variables in the model including demographic, exposure, and cleaning practices. All analyses were performed using SAS version 9.3 (SAS Institute Inc, Cary, NC, USA). 

## 3. Results

A total of 6447 lower Manhattan residents were included in this study. Table 1 shows their demographic characteristics. Mean age on 9/11 was 45.1 years (±15.1 years). Around 42% were male and 67% had completed college or post-graduate work. Forty-five percent had ever smoked cigarettes and 44% reported some or intense dust cloud exposure on 9/11. 

### 3.1. Geospatial Representation of Exposures and Symptoms

Figure 1, Figure 2 and Figure 3, respectively, display the geospatial distributions of self-reported outdoor location at the time of first encountering the dust cloud by intensity of exposure, indoor dust and damage to residences, and any reported respiratory symptoms up to 6 years after the 9/11 attacks. Figure 1 and Figure 2 show more frequent reports of higher indoor and outdoor exposures on edges of lower Manhattan, especially the western side along the Hudson River. Over 50% of WTCHR enrollees residing in census tracts in both the lower western and eastern sides of Manhattan reported having dust or debris in the home. The proportion reporting indoor home exposure declined rapidly further north and east up to the Registry Canal Street boundary for residents. The distribution of respiratory symptoms depicted in the Figure 3 map overlaps in large part with the distribution of reported residential exposures. The percentage of enrollees by census tract who reported any respiratory symptom ranged from less than 1% furthest North and East approaching Canal Street to 50% or more of enrollee respondents who lived in census tracts along the Hudson River adjacent to the WTC site.

### 3.2. Home Damage and Cleaning Practices

Table 2 shows the types of home damage and household cleaning practices of WTCHR lower Manhattan residents. The majority of residents reported a fine coating of dust on interior surfaces, while 17% had a heavy coating of dust, enough to make it impossible to see what was underneath. Only 6% had broken windows and less than 15% reported damage to home or furnishings. Presence of debris was reported by 12% of the respondents. Cleaning practices varied greatly: more than half of participants reported cleaning with a damp cloth, sponge, or mop, whereas 23% reported dusting or sweeping without water. Around 20% of participants replaced carpets, furniture, or drapes, blinds, and curtains. Almost one third replaced air conditioners (individually, or the building). 

Figure 2 shows that the majority of respondents reporting any type of home damage were located near the WTC site.

Table 3 shows that those who dusted or swept without water were more likely to report the presence of debris (aOR = 1.31; 95% Confidence Interval (CI) 1.06–1.61) as to other reported exposures.

Using a damp cloth, sponge, or mop was also associated with reporting a fine coating of dust (aOR = 1.85; 95% CI 1.59–2.14). In contrast, those who used water for cleaning were less likely to report broken windows. Lower Manhattan residents who reported using a vacuum to clean, as well as those who reported cleaning ventilation ducts, were less likely to report a fine or a heavy coating of dust, but more likely to report damage to home or furnishings, Table 3). Cleaning practices were also associated with each other, with cleaning with damp cloth, sponge, or mop having the strongest association with using a vacuum to clean (aOR = 4.71; 95% CI 4.04–5.50) and cleaning ventilation ducts (aOR = 5.28; 95% CI 3.72–7.50). These results were expected, yet including these terms made for more thorough modeling. 

Prevalence of respiratory outcomes was as follows: shortness of breath (16.1 %), wheezing (10.7%), chronic cough (6.8%), upper respiratory symptoms (60.8%), asthma or reactive airways dysfunction syndrome (RADS) (7.9%), and chronic obstructive pulmonary disease (COPD) (5.4%) (Table 4). A total of 68% of lower Manhattan residents reporting any respiratory outcome were located within 0.5 miles from Ground Zero (Figure 3). Table 5 shows odds ratios and 95% confidence intervals for respiratory outcomes in relation to several characteristics of home damage and cleaning practices. (See explanatory comments given previously, as Table 3 and Table 5 have the same layout). Those who reported a heavy coating of dust had 65% (95% CI 1.24–2.18), 43% (95% CI 1.03–1.97) and 59% (95% CI 1.09–2.28) higher odds of reporting shortness of breath, wheezing, or chronic cough, respectively. Residents who reported damage to home or furnishings had a 33% (95% CI 1.01–1.75) increased odds of reporting shortness of breath and 36% (95% CI 1.06–1.74) increased odds of reporting upper respiratory symptoms. Presence of debris was associated with chronic cough (aOR = 1.56; CI 1.12–2.17) and upper respiratory symptoms (aOR = 1.56; CI 1.24–1.95). Lower Manhattan residents who reported replacing air conditioners had higher odds of reporting upper respiratory symptoms (aOR = 1.32; CI 1.14–1.54). Replacing carpets was associated with increased COPD (aOR = 1.49; CI 1.03–2.16), while replacing drapes was associates with increased shortness of breath (aOR = 1.31; CI 1.03–1.65). Dusting or sweeping without water was the cleaning behavior associated with the largest number (three) of respiratory outcomes. Residents who did that had higher odds of reporting shortness of breath (aOR = 1.37; 95% CI 1.11–1.69), wheezing (aOR = 1.49; CI 1.17–1.90), and upper respiratory symptoms (aOR = 1.28; CI 1.08–1.53). Cleaning with a damp cloth, sponge, or mop and cleaning ventilation ducts were also associated with some respiratory outcomes including URS for damp cloth (aOR = 1.40; 95% CI 1.18–1.64) and wheezing and cough with ventilation duct cleaning (aOR = 1.48; 95% CI 1.08–2.01 and aOR = 1.94; 95% CI 1.38–2.73, respectively).

## 4. Discussion

This is the largest study to evaluate respiratory outcomes among lower Manhattan residents who reported household exposures as a result of the 9/11 terrorist attacks. Our findings are consistent with increased respiratory symptoms and diseases, which are associated with several levels of home damage and different cleaning practices, and corroborate other studies in similar populations [8,9,10,11,13] or subset of the WTCHR resident population [13].

The explosion and collapse of buildings and subsequent fires at Ground Zero produced a large plume of dust and smoke that released particles and gases into the air. Characterization of both outdoor and indoor dust samples identified asbestos, glass fibers, lead, and polycyclic aromatic hydrocarbons (PAHs), among numerous other contaminants [1,4,5]. In fact, indoor samples contained more inhalable dust particles than did outdoor dust [6]. Our data show that the majority of lower Manhattan residents in the WTCHR may have been exposed to airborne contaminants because of some type of home damage following the 9/11 attacks. The presence of dust was the most common problem, but more severe damage, such as broken windows and furniture, were also reported. A case-control study comparing lower and upper Manhattan residents found that 30.7% of residents in the affected region reported some physical damage to their homes and 86.4% reported dust [8]. As expected, our geospatial analysis demonstrates that a higher proportion of WTCHR participants reporting any home damage were located in the vicinity of Ground Zero.

As a result of the presence of dust, debris, and other damage, lower Manhattan residents personally cleaned their homes using a variety of methods. Lin et al. reported that 74.3% of residents in the affected area personally cleaned their homes [8]. The majority of residents participating in the present study reported using wet cleaning practices, such as using a damp cloth, sponge, or mop, which are likely to minimize suspension of dust and have been recommended by the Environmental Protection Agency (EPA) in that context [3,15]. However, many residents reported dusting or sweeping without water and even cleaning ventilation ducts by themselves. These practices were associated with the presence of debris and may have resulted in excessive dust exposure. In addition, numerous residents reported that household items, such as carpets, furniture, curtains, and air conditioners were replaced, which may have been associated with more severe damage to their homes. Among other measures, the EPA designed and implemented a Residential Dust Cleanup Program in 2002, to ensure that lower Manhattan residents were protected from potential WTC-related exposures [16].

Upper respiratory symptoms were the most common health outcomes reported by lower Manhattan residents in the WTCHR. This finding is consistent with other WTC studies among residents [10] and rescue and recovery workers [17,18,19]. According to Lin et al., residents living within 1.5 km of the WTC site experienced a 200 percent increase of at least one persistent upper respiratory symptom compared to controls in the Upper West Side of Manhattan [10]. Our study also demonstrates high rates of persistent shortness of breath, wheezing, and chronic cough among lower Manhattan residents. These findings are in line with those of Lin et al., who reported that increased rates of lower respiratory symptoms persisted among lower Manhattan residents two and four years after the 9/11 attacks [9]. Moreover, most residents reporting any persistent respiratory symptoms in our study were located within a 0.5-mile radius of Ground Zero. Lin et al. also described an apparent trend when comparing residents below and not below Canal St.: the former had stronger associations between location and both any new-onset and persistent new-onset respiratory symptoms [9]. Reibman et al. also reported that four times as many lower Manhattan residents presented wheezing compared to a control population. In addition, three times as many residents reported cough and shortness of breath [11].

Our results demonstrate an association between respiratory health outcomes and numerous types of home damage and cleaning practices. In particular, the presence of a heavy coating of dust was associated with shortness of breath, wheezing and cough and the presence of debris with chronic cough and upper respiratory symptoms. Lin et al. reported that upper respiratory symptoms were associated with both dust and physical damage to home, with statistically significant cumulative incidence ratios (CIRs) ranging from 1.27 to 1.71. In addition, the association between lower respiratory symptoms and exposures yielded CIRs between 1.31 and 1.44 [8]. In our study, the use of dry cleaning practices, such as dusting or sweeping without water, was associated with shortness of breath, wheezing, and upper respiratory symptoms. In contrast, only one health outcome was associated with cleaning with a damp cloth, sponge, or mop. It is not clear if the cleaning practice with water provided some protective effects by limiting re-suspension of dust, since in our analysis the cleaning practices with and without water were not mutually exclusive. Both practices were also associated with other methods of cleaning. Furthermore, cleaning practice with water was significantly associated with fine coating of dust (OR = 1.85; 95% CI, 1.59–2.14) versus cleaning practice without water was not (OR = 0.99; 95% CI, 0.82–1.20). Nevertheless, Lin et al. did not find statistically significant associations between self-cleaning of homes and lower respiratory symptoms at 2- and 4-year follow ups of lower Manhattan residents (N = 136 and 69 participants, respectively) [9]. 

A strength of this study is the ability to demonstrate persistent health outcomes in a large sample of lower Manhattan residents exposed in their homes to airborne contaminants resulting from the 9/11 terrorist attacks. Our analyses controlled for potential confounders, such as the intensity of exposure to the cloud of dust and debris, smoking, and priority group or area of recruitment, which may serve as a surrogate for location in lower Manhattan. 

However, this study is subject to several limitations. Self-selection bias may affect our results. Lower Manhattan residents who developed respiratory symptoms and diseases after 9/11/2001 may have been more likely to enroll in the WTC registry than asymptomatic persons. Medical records were not obtained which might have been used to verify reported conditions. In addition, smoking was defined as ever/never, without taking into account pack-years. Also, there was essentially no ability to validate the reported exposures quantitatively, because environmental sampling was not conducted. Nonetheless, it is unlikely that differences between respondents and non-respondents and misclassification of disease status would have an influence on the intensity of home damage reporting and selection of cleaning practices. Recall bias is also a limitation since questionnaires for W2 were applied over 5 years after the event of interest. In an attempt to account for this aspect, we verified responses on W1 and W2 for consistency. In addition, information on time to return home and use of respiratory protection while cleaning was not available. It is possible that severely damaged home inhabitants took longer to return to their homes or did so only after renovations, which potentially reduced dust exposures. Similarly, residents may have used different degrees of respiratory protection while cleaning surfaces. These factors would have *reduced* the apparent effect of the reported home damage on respiratory health outcomes.

## 5. Conclusions

This analysis demonstrates that lower Manhattan residents who suffered home damage following the 9/11 attacks are more likely to report respiratory symptoms and diseases in the WTCHR. It also highlights the specific kinds of damage or other specific exposure events, such as cleaning practices, which are statistically related to such increased symptoms and diseases. These health outcomes persisted for at least 5–6 years after the event, which may have translated into lower quality of life. 

## Figures and Tables

**Figure 1 ijerph-16-00798-f001:**
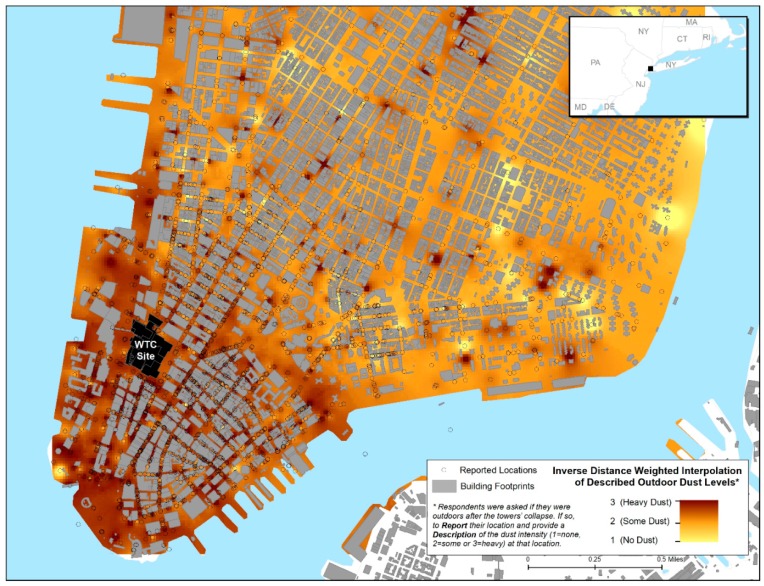
Inverse distance weighted (IDW) interpolation of reported levels of dust cloud intensity (1 = none, 2 = some or 3 = intense) among World Trade Center (WTC) Health Registry respondents (n = 23,466) (Wave 1 survey, 9/2003–11/2004).

**Figure 2 ijerph-16-00798-f002:**
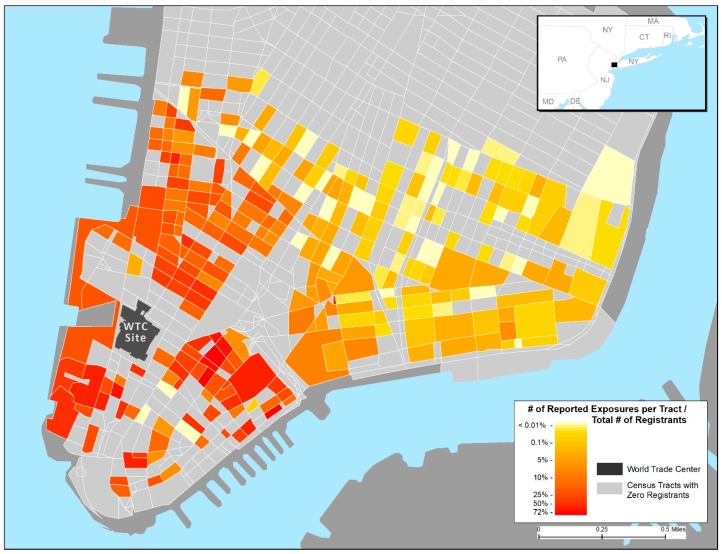
Proportion of World Trade Center (WTC) Health Registry participants reporting any fine coating of dust on surfaces, heavy coating of dust on surfaces, broken window (s), damage to home or furnishings, or debris in their Lower Manhattan residences (n = 6348) (Wave 2 survey, 11/2006–12/2007).

**Figure 3 ijerph-16-00798-f003:**
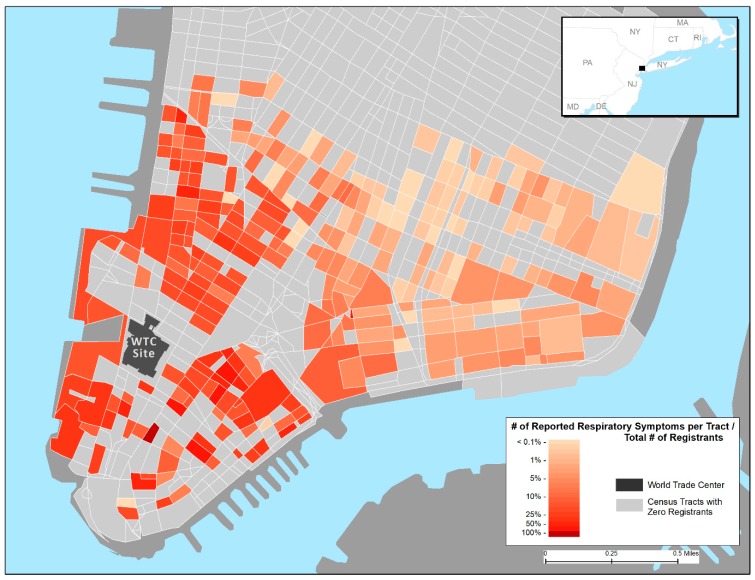
Proportion of World Trade Center (WTC) Health Registry participants reporting any respiratory outcome (shortness of breath, wheezing, chronic cough, upper respiratory symptoms, asthma/reactive airways dysfunction syndrome, chronic obstructive pulmonary disease) (n = 6348) (Wave 2 survey, 11/2006–12/2007).

**Table 1 ijerph-16-00798-t001:** Demographic characteristics of 6447 Lower Manhattan residents enrolled in the World Trade Center (WTC) Health Registry.

Characteristic	N	%
Age on 9/11/2001 (yrs, mean ± SD)	45.1 ± 15.1	-
Sex (male)	2688	41.7
Race		
White	4281	66.4
Black	325	5
Hispanic	579	9
Asian	1061	16.5
Other/Unknown	201	3.1
Education *		
<High school	504	7.9
High school	738	11.5
Some college	893	14
College +	4265	66.6
Income level		
Less than 25K	1322	20.9
25 to <50K	1181	18.6
50 to <75K	836	13.2
75 to <100K	1801	28.4
100K or more	1197	18.9
Ever smoking *	2903	45.4
Dust cloud exposure on 9/11 *		
None	3446	56.3
Some	1196	19.5
Intense	1479	24.2

* Percentages do not reflect total N due to missing values.

**Table 2 ijerph-16-00798-t002:** Self-reported household exposures resulting from the 9/11 attacks, cleaning practices, and other characteristics among 6447 Lower Manhattan residents enrolled in the World Trade Center (WTC) Health Registry.

Characteristics	N *	%
Fine coating of dust	3974/6241	63.7
Heavy coating of dust	1067/6251	17.1
Broken windows	368/6256	5.9
Damage to home or furnishings	927/6250	14.8
Presence of debris	759/6256	12.1
Cleaned ventilation ducts	1061/5931	17.9
Cleaned with damp cloth, sponge, or mop	3307/5919	55.9
Used a vacuum to clean	2687/5934	45.3
Dusted or swept without water	1263/5407	23.4
Replaced carpet or rugs	1201/6090	19.7
Replaced furniture	1304/6090	21.4
Replaced drapes, blinds, or curtains	1232/6090	20.2
Replaced air conditioners	1896/6090	31.1

* Changes in denominators are due to non-response.

**Table 3 ijerph-16-00798-t003:** Associations between cleaning practices and home exposures among 6,447 Lower Manhattan residents enrolled in the World Trade Center (WTC) Health Registry.

Exposure/Cleaning Practices	Adjusted Odds Ratio (95% Confidence Interval) *			
	Dusted or Swept without Water	Cleaned with Damp Cloth, Sponge, or Mop	Used a Vacuum to Clean	Cleaned Ventilation Ducts
Fine coating of dust	0.99 (0.82, 1.20)	**1.85 (1.59, 2.14)**	**0.63 (0.53, 0.74)**	**0.38 (0.27, 0.52)**
Heavy coating of dust	1.03 (0.82, 1.29)	1.08 (0.88, 1.33)	**0.64 (0.52, 0.79)**	**0.59 (0.42, 0.83)**
Broken windows	1.24 (0.86, 1.79)	**0.69 (0.48, 0.99)**	0.97 (0.67, 1.39)	0.95 (0.57, 1.59)
Damage to home or furnishings	0.92 (0.73, 1.14)	0.90 (0.72, 1.12)	**1.25 (1.01, 1.54)**	**1.34 (1.01, 1.79)**
Presence of debris	**1.31 (1.06, 1.61)**	1.07 (0.87, 1.33)	1.20 (0.98, 1.48)	1.16 (0.87, 1.53)
Dusted or swept without water	-	**1.50 (1.26, 1.79)**	**1.76 (1.50, 2.07)**	**1.76 (1.43, 2.18)**
Cleaned with damp cloth, sponge, or mop	**1.47 (1.23, 1.74)**	-	**4.59 (3.95, 5.34)**	**3.92 (2.94, 5.23)**
Used a vacuum to clean	**1.73 (1.48, 2.03)**	**4.71 (4.04, 5.50)**	-	**1.95 (1.56, 2.43)**
Cleaned ventilation ducts	**1.74 (1.41, 2.15)**	**5.28 (3.72, 7.50)**	**2.08 (1.65, 2.62)**	-

* Each model has terms for eight confounders: models are adjusted for any effect due to age at 9/11, education, income level, race, sex, smoking status, and dust cloud exposure and resident priority group. - = not in the particular logistic model. Each column gives a distinct logistic model fit, with the column heading being the dependent (response) cleaning practice variable. (Statistically significant adjusted odds ratios in bold).

**Table 4 ijerph-16-00798-t004:** Prevalence of self-reported post-9/11, respiratory symptoms and respiratory diseases present at Wave 2 (W2) (11/2006–12/2007) surveys among 6447 Lower Manhattan residents enrolled in the World Trade Center (WTC) Health Registry.

Symptom/Disease	N *	%
Shortness of breath	983/6126	16.1
Wheezing	668/6234	10.7
Chronic cough	439/6415	6.8
Upper respiratory symptoms	3761/6,190	60.8
Asthma/RADS ^†^	434/5466	7.9
Chronic obstructive pulmonary disease	320/5931	5.4

* Changes in denominators are due to non-response. ^†^ RADS = Reactive airways dysfunction syndrome.

**Table 5 ijerph-16-00798-t005:** Post-9/11 respiratory symptoms or diseases present at W2 in relation to home exposures and cleaning practices among 6447 Lower Manhattan residents enrolled in the World Trade Center (WTC) Health Registry.

Exposure/Cleaning Practices	Adjusted Odds Ratios (95% Confidence Interval) * for Each Respiratory Condition					
	Shortness of Breath	Wheezing	Chronic Cough	URS **	Asthma/RADS †	COPD ‡
Light coating of dust	1.25 (0.98–1.59)	1.10 (0.84, 1.45)	1.06 (0.77, 1.47)	1.11 (0.93, 1.31)	1.01 (0.74, 1.39)	1.19 (0.84, 1.70)
Broken Windows	0.95 (0.60, 1.50)	1.46 (0.91, 2.36)	0.88 (0.68, 1.62)	0.98 (0.66, 1.44)	1.16 (0.63, 2.12)	1.01 (0.51, 1.99)
Heavy coating of dust	**1.65 (1.24, 2.18)**	**1.43 (1.03, 1.97)**	**1.59 (1.09, 2.28)**	1.08 (0.87, 1.35)	1.21 (0.82, 1.79)	1.35 (0.89, 2.05)
Damage to home or furnishings	**1.33 (1.01, 1.75)**	1.31 (0.96–1.81)	0.82 (0.56, 1.19)	**1.36 (1.06, 1.74)**	1.11 (0.74, 1.65)	1.11 (0.73, 1.68)
Presence of debris	1.62 (0.97, 1.64)	1.19 (0.88, 1.61)	**1.56 (1.12, 2.17)**	**1.56 (1.24, 1.95)**	0.92 (0.63, 1.35)	0.94 (0.63, 1.40)
Replaced air conditioner	1.06 (0.87, 1.28)	1.08 (0.87, 1.34)	1.24 (0.96, 1.60)	**1.32 (1.14, 1.54)**	0.91 (0.69, 1.19)	1.31 (0.99, 1.74)
Replaced carpets	0.90 (0.70, 1.17)	1.05 (0.78, 1.41)	1.07 (0.76, 1.50)	1.03 (0.83, 1.28)	1.21 (0.85, 1.73)	1.49 (1.03, 2.16)
Replaced drapes	**1.31 (1.03, 1.65)**	0.98 (0.75, 1.29)	1.02 (0.74, 1.40)	1.11 (0.90, 1.36)	0.99 (0.71, 1.39)	0.96 (0.67, 1.38)
Replaced furniture	1.24 (0.97, 1.59)	1.13 (0.85, 1.51)	1.24 (0.89, 1.73)	1.14 (0.93, 1.40)	1.40 (0.99, 1.96)	0.89 (0.62, 1.29)
Dusted or swept without water	**1.37 (1.11, 1.69)**	**1.49 (1.17, 1.90)**	1.21 (0.91, 1.61)	**1.28 (1.08, 1.53)**	**1.35 (1.00, 1.81)**	1.16 (0.85, 1.61)
Cleaned with damp cloth, sponge, mop	1.15 (0.92, 1.43)	1.01 (0.79, 1.30)	1.05 (0.78, 1.41)	**1.40 (1.18, 1.64)**	0.91 (0.67, 1.22)	0.96 (0.69, 1.32)
Cleaned ventilation ducts	1.23 (0.93, 1.64)	**1.48 (1.08, 2.01)**	**1.94 (1.38, 2.73)**	1.18 (0.92, 1.53)	1.23 (0.82, 1.82)	1.30 (0.86, 1.96)
Vacuumed	0.96 (0.78, 1.19)	1.06 (0.84, 1.34)	0.95 (0.71, 1.25)	1.01 (0.86, 1.19)	0.99 (0.75, 1.33)	0.94 (0.69, 1.29)

* Each model has terms for eight confounders: models are adjusted for any effect due to age at 9/11, education, income level, race, sex, smoking status, and dust cloud exposure and resident priority group. ** URS = Upper respiratory symptoms. ^†^ RADS = Reactive airways dysfunction syndrome. ^‡^ COPD = Chronic obstructive pulmonary disease. Each column gives a separate logistic model fit, with the column heading being the dependent (response) health outcome variable. (Statistically significant adjusted odds ratios in bold).

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
