# Peer review of "9/11 Residential Exposures: The Impact of World Trade Center Dust on Respiratory Outcomes of Lower Manhattan Residents"

_ijerph, 2019, doi:10.3390/ijerph16050798_

Round 1
Reviewer 1 Report
This study assessed the association between respiratory outcomes and dust exposures due to 9/11 event. In general, the theme of the study fits the journal’s scope, and the results obtained from the study provide important insights to understand the impacts of disaster on exposure epidemiology and public health. In the followings please find my comments for the authors:
1. This study used questionnaire-based information to categorize subjects’ dust exposed level. Did you see any other studies related 9/11 but with different exposure assessment methodologies, ex. on-site monitoring or satellite measurements. Could you please briefly compare the differences between your paper and previous studies?
2. A binary term of the health outcomes coupled with a logistic model was used for the statistical analysis. This is fine, but I was still wondering is it possible to obtain the information like clinical visit for specific outcomes to further identify the effects of dust exposures on disease prevalence.
3. Do we need to adjust comorbidity during the model analysis?
4. I think it is worth to use the “distance to WTC” as a strata to conduct a subgroup analysis.
5. Why do you have two figure 3 (p9 and 10)? There are some format confusions on page 9. I cannot recognize whether links 258 to 259 are main text of a foot note belonged to a figure?
6. Although you have CI in Table 5, I would still suggest the authors to use the star sign to represent the significance to make it more intuitive.
Author Response
This study used questionnaire-based information to categorize subjects’ dust exposed level. Did you see any other studies related 9/11 but with different exposure assessment methodologies, ex. on-site monitoring or satellite measurements. Could you please briefly compare the differences between your paper and previous studies?
The authors are not aware of publications that have directly correlated actual airborne dust measurements and health outcomes. Most studies after 9/11 used self-reported exposures and symptoms and were included in our discussion. No change in manuscript.
A binary term of the health outcomes coupled with a logistic model was used for the statistical analysis. This is fine, but I was still wondering is it possible to obtain the information like clinical visit for specific outcomes to further identify the effects of dust exposures on disease prevalence.
Unfortunately, medical records or discharge information were not available to perform the suggest analyses. No change in manuscript.
Do we need to adjust comorbidity during the model analysis?
All analyses were adjusted for potential confounders. There were no relevant comorbidities to adjust for in the respiratory health analyses. No change in manuscript.
.I think it is worth to use the “distance to WTC” as a strata to conduct a subgroup analysis.
We did try to use distance from WTC in a preliminary analysis, but found that Priority Group was a better surrogate for location. No change in manuscript.
. Why do you have two figure 3 (p9 and 10)? There are some format confusions on page 9. I cannot recognize whether links 258 to 259 are main text of a foot note belonged to a figure?The duplicate figure might have been inserted by mistake by the journal typesetters.
This was fixed.
Although you have CI in Table 5, I would still suggest the authors to use the star sign to represent the significance to make it more intuitive.
The statistically significant Cis are bolded, per journal’s style. No change in manuscript.
Reviewer 2 Report
The authors present an interesting investigation on the association between exposures and cleaning practices after 9/11 and respiratory outcomes in residents of lower Manhattan. Overall, the methods are sound and the presentation of the results are clear. I have the following comments:
- I have an issue with the changing N of the participants included in the analyses. The authors cite 6447 in their main analyses but based on the descriptive tables, it seems that this number is not correct. The self-reported household exposure responses range from 5931 to 6256. the maps for symptoms cite N=6348 while there are different Ns for the report of symptoms in Table IV. The true Ns for each logistic regression presented should be included in the tables or the authors should state if they any data imputation.
- How do the authors explain the statistically significant ORs that are less than 1 in table III, ie fine coating of dust and cleaned ventilation ducts.
- it would be beneficial to include a bit of mechanistic support in the discussion as to how the dust cloud exposure is associated with chronic respiratory illness.
Minor comments:
-typos in page 3 line 95, page 5 line 151 and 159
- the row alignment for income level is off in Table I
-Table 5 COPD column is not aligned.
Author Response
IJERPH-450489
Response to reviewers
The authors present an interesting investigation on the association between exposures and cleaning practices after 9/11 and respiratory outcomes in residents of lower Manhattan. Overall, the methods are sound and the presentation of the results are clear. I have the following comments:
I have an issue with the changing N of the participants included in the analyses. The authors cite 6447 in their main analyses but based on the descriptive tables, it seems that this number is not correct. The self-reported household exposure responses range from 5931 to 6256. the maps for symptoms cite N=6348 while there are different Ns for the report of symptoms in Table IV. The true Ns for each logistic regression presented should be included in the tables or the authors should state if they any data imputation.
The changing N is due to missing values in the different analyses, as stated in the manuscript. No imputation was performed.
- How do the authors explain the statistically significant ORs that are less than 1 in table III, ie fine coating of dust and cleaned ventilation ducts.
The ORs merely reflect cleaning practices adopted by lower Manhattan residents. Unfortunately, the authors cannot explain why those who used water for cleaning were less likely to report broken windows or those who reported using a vacuum to clean, as well as those who reported cleaning ventilation ducts, were less likely to report a fine or a heavy coating of dust.
it would be beneficial to include a bit of mechanistic support in the discussion as to how the dust cloud exposure is associated with chronic respiratory illness.
The fact that many substances found in the dust cloud, such as cement dust, glass fibers, asbestos, polycyclic aromatic hydrocarbons, polychlorinated biphenyls, organochlorine pesticides, and polychlorinated furans and dioxins cause respiratory illness is well known. Nevertheless, the authors decided to refrain from a more mechanistic approach given the nature of the self-reported data and stayed in the ‘associations’ realm.
-typos in page 3 line 95, page 5 line 151 and 159
Typos were corrected.
the row alignment for income level is off in Table I
Row alignment corrected.
Table 5 COPD column is not aligned.